# Comparative Analyses of Four Chemicals Used to Control Black Mold Disease in Tomato and Its Effects on Defense Signaling Pathways, Productivity and Quality Traits

**DOI:** 10.3390/plants9070808

**Published:** 2020-06-28

**Authors:** Hoda A. S. El-Garhy, Fayz A. Abdel-Rahman, Abdelhakeem S. Shams, Gamal H. Osman, Mahmoud M. A. Moustafa

**Affiliations:** 1Genetics and Genetic Engineering Dept., Faculty of Agriculture, Benha University, Qalyubia 13736, Egypt; Hoda.algarhy@fagr.bu.edu.eg (H.A.S.E.-G.); mahmoud.mustafa@fagr.bu.edu.eg (M.M.A.M.); 2Postharvest Diseases Dept., Plant Pathology Research Institute, ARC, Giza 12619, Egypt; Fayz17102@gmail.com; 3Horticulture Dept., Faculty of Agriculture, Benha University, Qalyubia 13736, Egypt; abdelhakeem.shams@fagr.bu.edu.eg; 4Department of Biology, Faculty of Applied Sciences, Umm Al-Qura University, Makkah 21955, Saudi Arabia; 5Research Laboratories Center, Faculty of Applied Science, Umm Al-Qura University, Makkah 21955, Saudi Arabia; 6Agricultural Genetic Engineering Research Institute (AGERI), ARC, Giza 12619, Egypt

**Keywords:** safe chemical inducers, foliar application, defense genes, *Alternaria alternata*, tomato black mold disease

## Abstract

The field application of safe chemical inducers plays a vital role in the stimulation of systematic acquired resistance (SAR) of plants. In this study, the efficacy use of three and six field applications with chitosan, lithovit, and K-thiosulfate at 4 g L^−1^ and salicylic acid at 1.5 g L^−1^ in improving tomato productivity, quality, and modifying the defense signaling pathways to the *Alternaria alternata* infection was investigated. Salicylic acid was the most effective in vitro where it completely inhibited the growth of *Alternaria alternata*. The highest yield quantity was recorded with six applications with Chitosan followed by Salicylic acid; also, they were the most effective treatments in controlling the *Alternaria alternata* infection in tomato fruits. The maximum increase in chitinase and catalase activity of tomato fruits was observed at five days after inoculation, following treatment with six sprays of salicylic acid followed by chitosan. The transcript levels of seven defense-related genes: ethylene-responsive transcription factor 3 (*RAP*), xyloglucan endotransglucosylase 2 (*XET-2*), catalytic hydrolase -2 (*ACS-2*), proteinase inhibitor II (*PINII*), phenylalanine ammonia-lyase 5 (*PAL5*), lipoxygenase D (*LOXD*), and pathogenesis-related protein 1 (*PR1*) were upregulated in response to all treatments. The highest expression levels of the seven studied genes were recorded in response to six foliar applications with chitosan. Chitosan followed by salicylic acid was the most effective among the tested elicitors in controlling the black mold rot in tomato fruits. In conclusion, pre-harvest chitosan and salicylic acid in vivo application with six sprays could be recommended as effective safe alternatives to fungicides against black mold disease in tomato fruits.

## 1. Introduction

Tomato (*Lycopersicon esculentum*, Mill) is an important vegetable crop in several markets around the world [1]. It contains different classes of antioxidants, such as carotenoids, ascorbic acid, phenolic compounds and α-tocopherol [2]. However, tomato production is threatened by many different pathogens that affect quality and productivity [3].

The rotting of tomato fruits by fungi is observed during the various production stages in the field, storage, transport, and marketing. Black mold of tomato fruits, caused by *Alternaria alternata*, is among the most common diseases, causing significant post-harvest losses in the quality of fruits, rendering large amounts of tomato fruits unfit for marketing and consumption [4]. 

Induction of resistance in tomato fruits could effectively control the black mold rot of tomato fruits caused by *Alternaria alternata*. Chitosan is an easily available and renewable natural polymer that stimulates plant growth [5]. Moreover, it exhibits antifungal effects against a wide range of fungi, including *Alternaria alternata*, *Botrytis cinerea*, *Rhizopus stolonifera*, and *P. expansum* [6]. These positive effects of chitosan can be attributed to the elicitation of phytoalexins and defense barriers in the host tissues [7]. Furthermore, coating tomato fruits reduces the severity of infection with pathogens and improves the storability of postharvest fruits and vegetables [8].

Salicylic acid (SA) is an organic acid that is involved in the regulation of various physiological processes in plants [9]. In systemic acquired resistance (SAR), it plays a central role in the signaling pathways [10]. Thus, spraying tomato plants with SA increased the activities of superoxide dismutase (SOD), peroxidase (POD), and catalase (CAT) in the leaves, as well as ascorbate peroxidase and glutathione reductase [11]. Additionally, the application of SA had no negative implications on biomass accumulation, and increased the postharvest life of tomato fruits [12].

Lithovit is a foliar fertilizer that improves plant growth and the yield of common beans [13], sugar beet [14], snap bean [15], and melon [16]. Lithovit contains several elements or compounds, such as magnesium, which is the central element of the chlorophyll molecule; silica, which plays a vital role in increasing plant tolerance against abiotic stress, mainly with respect to water relations and photosynthesis [17]; and calcium carbonate (CaCO_3_), which is decomposed to calcium oxide (CaO) and carbon dioxide (CO_2_) in the leaf stomata, where CO_2_ increases the intensity of photosynthesis [18]. It has been proposed that plant resistance to pathogens and pests can be increased by field application of these chemical elicitors [19].

Potassium (K^+^) is a macronutrient required for fundamental physiological and molecular processes in plants [20]. It is probably involved in the development of stronger cell walls for preventing the entry of plant pathogens into the cell [21]. Regarding this, the pre-harvest spraying of bean plants with potassium thiosulfate was shown to significantly decrease the pod rot caused by *B. cinerea* and *Sclerotinia sclerotiorum* during storage [22].

Systemic acquired resistance (SAR) is a mechanism by which plant defenses are preconditioned by a prior treatment that results in enhanced resistance against subsequent pathogen infections. Although the molecular details of the signaling machinery are poorly understood [23], previous studies have shown that elicitation by biotic and abiotic stresses has a regulatory role in the expression of ethylene-responsive transcription factor 3 (*RAP*), xyloglucan endotransglucosylase 2 (*XET–2*), catalytic hydrolase–2 (*ACS–2*), proteinase inhibitor II (*PINII*), phenylalanine ammonia-lyase 5 (*PAL5*), lipoxygenase D (*LOXD*), and pathogenesis-related protein 1 (*PR1*) genes. These seven defense genes belonged to ethylene, jasmonic acid, and salicylic acid pathways, which are also the mechanisms of plant SAR [24,25,26].

This study aimed to analyze the effectiveness of chitosan, salicylic acid, lithovit, and potassium thiosulfate in improving tomato productivity and quality. We assessed if these chemicals affect fungal growth, plant growth parameters, cold storage period, yield components, and fruit quality, as well as modifying post-harvest resistance to *Alternaria alternata* infection. Finally, we addressed whether we could identify the induction of SAR-related defense genes by analyzing the expression of *RAP*, *XET–2*, *ACS–2*, *PINII*, *PAL5*, *LOXD*, and *PR1*, which are expressed as the key genes of ethylene, jasmonate, and salicylic acid pathways.

## 2. Materials and Methods

### 2.1. Source of Chemicals Tested as Inducers

Potassium thiosulfate, salicylic acid, and chitosan were procured from Sigma Chemical Co. (St. Louis, MO, USA), while lithovit was purchased from Zeovita, GmbH, Germany. A stock solution of chitosan (10 mg/mL) was prepared by dissolving 2 g of purified chitosan powder in 100 mL of distilled water containing 2 mL of acetic acid (stirred for 24 h), and the volume was made up to 200 mL with distilled water [27]. Concentrations of 2 and 4 mg/mL were then made by appropriate dilution. The pH of the solution was adjusted to 5.6 using 1N NaOH and sterilized by autoclaving at 121 °C for 15 min [28]. Lithovit^®^ is the commercial name of a natural organic calcite carbonate that is mined from natural limestone deposits and used as a foliar fertilizer. It is produced in Germany and distributed by Filmchem Ltd. Lithovit spraying solution was prepared by adding 4 g of the solid material per liter of water. Lithovit (micronized calcium carbonate) contains 75% calcium carbonate, 4% magnesium carbonate, 0.5% iron, 5% silica, 0.1% potassium oxide, 0.015% sodium, 0.015% phosphate, and 0.01% manganese.

### 2.2. Soil Analysis

Before transplanting the tomato seedlings, soil samples (0–30 cm depth) were collected from the experimental farm of Fac. Agric., Moshtohor, Benha University, Qalyubia Governorate, Egypt. The physical and chemical properties of these soil samples were analyzed as described by Jackson [29]. The physical and chemical properties of the investigated soil are shown in Table 1.

### 2.3. Fungal Isolates

*Alternaria alternata*, the causative agent of black mold rot on tomato fruits, was isolated from diseased tomato fruits obtained from storage refrigerators at the Giza governorate. The purified isolate was identified according to its morphological features, as described by Barnett and Hunter (1972) [30]. Identification of the isolated fungal strain was confirmed at the Mycology Research and Disease Survey Dept., Plant Pathology Research Institute, ARC, Giza governorate, Egypt. The isolate was cultured on potato dextrose agar (PDA) medium in Petri dishes for seven days at 25 ± 2 °C for further in vitro studies, as well as for the artificial inoculation of tomato fruits in disease control experiments.

### 2.4. Evaluation of Antifungal Activity In Vitro

The effect of the chemicals (chitosan, lithovit, potassium thiosulfate, and salicylic acid) on the growth of *Alternaria alternata* was studied in vitro. For this, 100 mL PDA was prepared in 250 mL conical flasks and autoclaved at 120 °C for 20 min. Chitosan, lithovit, potassium thiosulfate, and salicylic acid were added to the sterilized PDA medium at concentrations of 2 and 4 mg/mL, while salicylic acid was used at concentrations of 0.75 and 1.5 mg/mL. Ten milliliters of medium modified or non-modified with these four tested chemicals were poured into three Petri dishes as replicates for each treatment. Equal-sized discs (diameter of 3 mm) of seven-day-old cultures of *Alternaria alternata* were placed centrally onto the surface of solidified PDA plates, and then incubated at 25 ± 2 °C for seven days. The diameters of developed colonies were measured when fungal mycelia covered one plate in the control treatment, and the percentage of reduction in the colony diameter was calculated using the formula suggested by Sirirat et al. [31] as follows:(1)$$\%\;\mathrm{Efficacy}=\frac{\Delta do-\Delta d}{\Delta do}\times 100$$
where, Δ*do* and Δ*d* were the average diameters of the fungal colonies in the control and treatment sets, respectively.

### 2.5. Field Studies (In Vivo)

The field experiment was carried out using tomato plants (*Lycopersicon esculentum* Mill cv. Super Strain B) at the experimental farm of the Faculty of Agriculture, Moshtohor, Benha University, Qalyubia, Egypt, during two successive seasons of 2018 and 2019. The field experiment was designed as a split plot with three replicates. Each plot (area 10 m^2^) consisted of four rows (each row being 3.75 m long and 0.7 m apart, 0.25 m deep, and having 12 hills located 0.30 cm apart. The chosen tomato seedlings were transplanted on one side of the row. Fertilizers were added in the form of nitrogen (NH_4_NO_3_, 33.5% N), phosphorus (Ca (H_2_PO_4_)_2_, CaCO_3_, 16% P_2_O_5_), and potassium (K_2_SO_4_, 48% K_2_O) to all the plots at rates of 335 kg N/ha^−1^, 153 kg P_2_O_5_/ha^−1^, and 115 kg K_2_O/ha^−1^, respectively. The experiment was divided into two groups. In the first group, the chemical inducers (chitosan, lithovet, potassium thiosulfate, and salicylic acid) and distilled water as the control were sprayed six times onto grown tomato plants: at three weeks from transplantation and subsequently at intervals of two weeks between sprays. In the second group, tomato plants were sprayed three times: at bloom stage and subsequently at intervals of two weeks between sprays. Chitosan, lithovet, and potassium thiosulfate were sprayed at concentrations of 4 g L^−1^, and salicylic acid at 1.5 g L^−1^.

### 2.6. Effect of the Pre-Harvest Foliar Application of Tested Chemicals on Some Growth Parameters, Yield, and Quality of Tomato Fruits

The effect of spraying tomato plants with the tested chemicals under field conditions during 2018 and 2019 was determined by recording the vegetative growth parameters, fruit yield, and quality of obtained tomato fruits, as follows:

#### 2.6.1. Vegetative Growth Characters

Five plants were taken randomly from each plot (100 days post transplantation), following which the plant height, leaf area per plant, as well as the fresh and dry weights per plant were recorded.

#### 2.6.2. Fruit Yield and Their Components

All the harvested fruits from each plot during the growing seasons were weighed and the total fruit yield was calculated. Additionally, the total yield per plant and hectare were estimated as an average of fruit weight.

#### 2.6.3. Fruit Quality

Levels of vitamin C and acidity were determined according to A.O.A.C. [32].

### 2.7. Effect of Tested Chemicals as a Pre-Harvest Foliar Application on Tomato Black Mold Rot During Storage under Cold Conditions

Seven days following the final application of the tested chemicals, tomato fruits from each treatment were harvested and transferred to the Post-Harvest Diseases Department, Plant Pathology Institute, ARC, Giza, Egypt. Tomato fruits were graded based on uniformity of size and maturity (pink stage of red color development). A fresh sample of the tomato fruits was divided into two groups. In the first group, tomato fruits were surface disinfected by dipping them into 2% sodium hypochlorite solution for 2 min, rinsed three times with sterilized distilled water, and left to air dry onto filter paper, prior to use. The sterilized tomato fruits were then inoculated with the prepared spore suspension of *Alternaria alternata* using an atomizer, which was prepared by brushing the surface of the culture in the presence of 10 mL of sterilized water per Petri-plate (diameter of 9 cm) and then filtering the spore suspension through nylon mesh. The concentration of spore suspension was adjusted to 3 × 10^5^ conidia/mL using a hemocytometer. In the second group, tomato fruits were left without sterilization (natural infection). Thereafter, all the treated fruits were air dried, placed into carton boxes (15 fruits per box), and stored in a cold room at 10 ± 1 °C and 90–95% RH for 7 and 15 days. Three boxes as replicates were used for each treatment as well as the control (sprayed tomato plants with distilled water). This experiment was carried out as a complete randomized design (CRD) with three replicates. The disease severity was calculated using the following formula described by Romanazzi et al. [33]:(2)$$\%\;\mathrm{Disease}\;\mathrm{Severity}=\frac{\sum \left(d\;\times \;f\right)}{N\;\times \;D}\;\times \;100$$
where d was the category of rot intensity scored on the fruit and f is its frequency, N is the total number of fruits examined (healthy and infected), and D is the highest category of decay intensity occurring on the empirical scale, using the six-degree scale, as 0 = healthy fruit, 1 = decayed area of the fruit ranging from 1% to 20%, 2 = decayed area of the fruit ranging from 21% to 40%, 3 = decayed area of the fruit ranging from 41% to 60%, 4 = decayed area of the fruit ranging from 61% to 80%, and 5 = at least 81% of the fruit surface is infected and showing sporulation.

### 2.8. Effect of Pre-Harvest Foliar Application of the Tested Chemicals on the Activities of Catalase and Chitinase in Treated Tomato Fruits

#### 2.8.1. Enzyme Extraction

Tomato fruits that were naturally or artificially infected with black mold rot were used to determine the activities of catalase and chitinase enzymes produced in vivo conditions at two and five days post inoculation. The fruit tissues collected from each treatment were homogenized in liquid nitrogen immediately. One gram of the powdered samples was extracted with 2 mL of 0.1 M sodium phosphate buffer (pH 7.0) at 4 °C. The homogenate was centrifuged at 4000 rpm at 4 °C for 20 min. The curd extract was used for determining the activities of catalase and chitinase [34].

#### 2.8.2. Chitinase Activity

The chitinase activity was measured according to the method described by Miller [35]. The activity of chitinase was expressed as µmoles of GlcNAc equivalents min^−1^ g^−1^ fresh weight at 575nm using a spectrophotometer (Spectronic 601 Milton ROY).

#### 2.8.3. Catalase Activity

The enzyme catalase was assayed according to the method described by Kato and Shimizu [36]. Catalase activity was calculated as the change in absorbency of the mixture min^−1^ at 240 nm using a spectrophotometer (Spectronic 601 Milton ROY).

### 2.9. Effect of the Pre-Harvest Foliar Application of Tested Chemicals on the Expression Signals of Resistance Genes in Treated Tomato Plants

#### 2.9.1. Extraction of RNA and cDNA Synthesis

Young tomato leaf samples were collected after three and six applications with the chemicals at the concentrations described above. They were transferred immediately to the Biotechnology Lab II, Genetics and Genetic Engineering Dept., Faculty of Agriculture, Benha University, and then homogenized in liquid nitrogen. The total RNA was extracted from the tomato leaf tissues using RNeasy^®^ Plant Mini kit (Qiagen, Cat. no. 51, 304), according to the manufacturer’s protocol. The RNA samples that were contaminated with gDNA were treated with gDNA Wipeout Buffer (QuantiTect^®^ Reverse Transcription Kit) according to the manufacturer’s instructions. The treated RNA samples were subjected to QuantiTect^®^ Reverse Transcription Kit (Qiagen, Cat. No. 205, 311) to obtain complementary DNA (cDNA), which were then stored at −20 °C until further investigation.

#### 2.9.2. Gene Expression Analysis Using qRT-PCR

The real-time quantitative reverse transcription-polymerase chain reaction (qRT-PCR) was performed in triplicate for each cDNA sample, as well as the negative and positive cDNA template controls. Each qRT-PCR reaction comprised of 2.5 µL of cDNA, 12.5 µL of SYBR Green PCR Master Mix (QuantiTect SYBR Green PCR Kit, Qiagen Cat. no. 204, 143), 0.3 µM of each of the forward and reverse primers presented in Table 2, 1 µL of RNase inhibitor, and RNase-free water to a final volume of 25 µL. The reactions were then analyzed on an AriaMx Real-Time PCR System (Agilent technologies) under the following conditions: 95 °C for 10 min, followed by 45 cycles of 95 °C for 20 s, 60 °C for 20 s, and 72 °C for 20 s. The fluorescence was monitored at the end of each cycle and finally, a temperature of 95 °C was maintained for 15 min for analysis of the melting temperature. Actin gene was used as an internal reference gene for qRT-PCR data normalization, since its expression remained more stable during the experiments [24]. All the experimentally induced changes in the expression of the studied genes were presented as n-fold changes relative to the corresponding controls. Relative gene expression ratios (RQ) between treated and control groups were calculated using the formula: RQ = 2^−ΔΔCT^ [37].

### 2.10. Statistical Analysis

The data were statistically analyzed using CoStat©3.4 software. The experiments of growth characters, yield, and quality of tomato fruits were conducted in a split plot design, with three replicates for each treatment. The in vitro and in vivo studies to assess the efficacy of the tested chemical inducers on *A. alternata* were designed using a completely randomized design with three replicates for each treatment. The means were compared using Duncan’s multiple range test at *p* < 0.05.

## 3. Results and Discussion

### 3.1. Effect of the Tested Chemicals on the Growth of Alternaria Alternata In Vitro

Four compounds, salicylic acid, potassium thiosulfate, chitosan, and lithovet were tested at different concentrations for their effects on the growth of *Alternaria alternata*. Results in Table 3 indicated that all the evaluated chemicals at varying concentrations reduced the linear growth of *A. alternata* compared to the control. In this respect, salicylic acid was the most effective among all the tested chemicals, as its tested concentration completely inhibited the growth of *A. alternata*. Additionally, increasing the concentration of potassium thiosulfate, chitosan, and lithovit showed a dosage response by inhibiting the growth of the tested pathogens. Moreover, the mycelial growth of the tested pathogen was more sensitive to high concentrations of potassium thiosulfate, compared to chitosan and lithovit. These results are in line with the findings of Abdel-Mageed et al. [38] who reported that salicylic acid completely inhibited the growth of *B. cinerea* at all the tested concentrations. Moreover, the modification of media by salicylic acid completely inhibited the linear growth of *A. solani* [39], as well as a complete reduction in the growth of *Fusarium mangiferae* [40]. On the other hand, El-Garhy et al. [22] observed that potassium thiosulfate (KTS) moderately suppressed the growth of *B. cinerea* and *P. aphanidermatum* in vitro. Chitosan interferes with the fungal membrane function, causing alterations in the permeability of the membrane and promoting internal osmotic imbalance. [41], reported that the mycelial growth of *A. alternata* was significantly reduced on media modified by the tested chitosan. Chitosan also considerably reduced the growth of *Botrytis cinerea* in vitro [42].

### 3.2. Field Studies (In Vivo)

#### Effect of the Pre-Harvest Foliar Application of Tested Chemicals on the Vegetative Growth Characters of Tomato Plants

Foliar application of the tested chemicals—chitosan, lithovit, potassium thiosulfate, and salicylic acid—on tomato plants improved the vegetative growth parameters of the plants, such as plant height, leaf area, and fresh and dry weight, compared to the control (plants sprayed with distilled water) during two successive seasons (Figure 1). Moreover, increasing the number of foliar sprays (six times) had a significant positive effect on these growth parameters. Furthermore, no significant variations were observed in plant heights in response to either three times or six times of foliar spray during the two successive seasons (Figure 1A). The largest leaf area was recorded with chitosan treatment, especially when sprayed six times rather than three times (Figure 1B). This was followed by salicylic acid. Likewise, spraying tomato plants with chitosan or salicylic acid resulted in the highest increase in the plant fresh and dry weight, especially after six applications (Figure 1C,D, respectively). These findings agree with those reported by Khandaker et al. [43] who observed significant improvements in the plant height, stem length, number and size of leaves, chlorophyll content, and antioxidant activity in red amaranth plants sprayed with salicylic acid. Additionally, Shams and Morsy [5] observed significant increases in the height and number of leaves in tomato plants, as well as the fresh and dry weight of the different plant organs, as a result of foliar application of chitosan. Moreover, foliar spray with lithovit significantly improved the growth parameters of broccoli, [44] tomato [18], and melon. Our results could also be supported by the findings of [45], who demonstrated the positive effect of salicylic acid as a growth regulator that improved plant growth, induced flowering, and increased the cellular metabolic rate [41] and also increased the tolerance to heat, chilling, and drought stresses [46]. On the other hand, chitosan can reduce the rate of transpiration through the formation of a thin anti-transpiring film over the leaves [47], in addition to its ability to control the pre- and post-harvest diseases [48]. Lithovit contains a balanced content of nutrients [49] such as Mg^++^, which is involved in the formation of building units of chlorophyll [50]. Furthermore, its finely sprayed particles can be absorbed into leaves and changed into CO_2_, thus enhancing the rate of photosynthesis [19]. It is well known that K^+^ is an essential nutrient for plant growth and yield [50]. K^+^ is not freely available for plant uptake in clay soil where its availability depends on soil properties [51]. Alternatively, the foliar spray of K^+^ fertilizers can significantly improve its concentrations in plants and consequently increase plant growth parameters, and therefore the yield outcome [52].

### 3.3. Effect of the Pre-Harvest Foliar Application of the Tested Chemicals on Yield and Quality of Tomato Fruits

As observed in Figure 2, the foliar application of chitosan or salicylic acid increased the weight of tomato fruits (Figure 2A), as well as the total fruit production (Figure 2B). Such increases exceeded those achieved from the lithovit and K-thiosulfate treatments. In general, all the investigated foliar applications increased the vitamin C content in tomato fruits, while chitosan and salicylic acid effected the highest increase in vitamin C in treated tomato fruits, followed by lithovit and K-thiosulfate (Figure 2C). As for the acidity of the tomato fruits, salicylic acid significantly increased the acidity to values that were higher than those caused by the other treatments (Figure 2D). Additionally, by increasing the number of sprays of all the tested chemicals from three to six sprays, the quantity and quality of harvested tomato fruits increased. These results agree with the findings of Sharif et al [53] who suggested that chitosan is a polysaccharide-based biopolymer that stimulates the activity of plant symbiotic microbes. Likewise, salicylic acid is known to reduce oxidative damage caused by environmental stresses, such as chilling and salinity [54], and improve the yield and yield attributes of cherry tomatoes [55]. In the case of lithovit, this modification provides plants with Ca^++^ in a micronized form that can be easily absorbed by plants [56]. This might improve plant metabolism [57]. The K-fertilizer is another factor strongly affecting the attributes that determine the fruit marketability [49].

### 3.4. Effect of the Pre-Harvest Foliar Application of the Tested Chemicals on Tomato Black Rot during Storage under Cold Conditions

The effect of the tested compounds on tomato black rot development during storage under cold conditions is presented in Table 4 and Table 5. It was observed that increasing the number of foliar application sprays from three to six sprays had a positive effect on reducing tomato black rot infection. All the tested chemicals significantly reduced the disease severity of *A. alternata* on tomato fruits, compared to the control. Six sprays of chitosan or salicylic acid were observed to be the most potent treatments as they completely suppressed black mold rot induced by *A. alternata* in both naturally infected or artificially inoculated tomato fruits that were cold-stored for seven days during the two successive seasons. Moreover, spraying tomato plants three times with salicylic acid completely inhibited the black mold infection in artificially inoculated tomato fruits post harvesting at seven days in cold storage, with efficacy reaching 100% during two successive seasons. These results are supported by the findings of Shafiee [58], who reported that salicylic acid completely inhibited natural fruit rot caused by pre-harvest latent infection or post-harvest inoculation. Salicylic acid completely inhibited the disease infection and lesion area in jujube fruits inoculated with *A. alternata* [59]. These results also agree with those of Mazaro et al. [60], who reported that strawberry fruits from plants treated with chitosan in the field at the rate of 0.5%, 1%, and 2% showed a reduced incidence of post-harvest rot. El-Garhy et al. [22] demonstrated that the pre-harvest spraying of bean plants with potassium thiosulfate decreased the pod rots caused by *Botrytis cinerea* and *Sclerotinia sclerotiorum*. Additionally, chitosan significantly decreased the microbial population on banana fruits during storage [61]. In the case of 15 days of cold storage, spraying tomato plants six times with chitosan was significantly more effective in reducing black mold infection in tomato fruits inoculated with *A. alternata* compared to three sprays. Similar results were observed with the application of salicylic acid, potassium thiosulfate, and lithovit. In this respect, spraying tomato plants six times with salicylic acid was highly effective in controlling the infection of tomato fruits with *A. alternata*, having an efficacy of reducing more than 90.0% of the disease severity. The next effective treatment was three sprays of salicylic acid that had an efficacy of reducing 80% of the disease severity, followed by lithovit. Potassium thiosulfate was the least effective in reducing the disease severity of *A. alternata* in tomato fruits. These results are in accordance with the findings of Feliziani et al [6] who observed that the pre-harvest or post-harvest application of chitosan was the most effective in reducing the storage rot of sweet cherry fruits.

Additionally, water-soluble chitosan (WSC) inhibited the development of anthracnose based on the diameter of the lesion in chili pepper fruits that were inoculated with *Colletotrichum capsici* [62]. Foliar application of tomato plants with salicylic acid enhanced their resistance to fruit rots caused by *Alternaria solani*, with the treatment decreasing the development of post-harvest fruit rot disease [63]. As for naturally infected tomato fruits, only six sprays of all the tested chemicals, except potassium thiosulfate, totally inhibited the development of black mold infection in tomato fruits during cold storage for 15 days during the two seasons. Among the tested chemicals, salicylic acid was the most effective in controlling the black mold infection, followed by chitosan, potassium thiosulfate, and lithovit. Generally, all the tested chemicals completely inhibited the development of black mold rot caused by *A. alternata* in naturally infected tomato fruits during cold storage for seven days, during the two seasons. These results agree with those recorded by Jiankang et al. [59] who reported that salicylic acid significantly reduced the decay rate in naturally infected jujubes during cold storage, and increased the activity of the main defense-related enzymes, including peroxidase, phenylalanine ammonia lyase, 1, 3-glucanase, and chitinase β in fruits during storage. Additionally, the pre-harvest spraying of bean plants with potassium thiosulfate and salicylic acid was more effective in decreasing the grey mold and cotton rot caused by *B. cinerea* and *Pythium aphanidermatum* in both naturally infected or artificially inoculated bean pods during cold storage [64]. In general, among all tested treatments there were no statistical differences, especially after seven days cold storage.

### 3.5. Effect of the Pre-Harvest Foliar Application of the Tested Chemicals on Activities of Chitinase and Catalase in Treated Tomato Fruits

The increasing of chitinase and catalase activities was associated with increasing resistance against infection by many fungal diseases and related to a greater tolerance to oxidative damage resulted from cold storage as abiotic stress. All four tested chemicals positively influenced the activities of chitinase in fruits of treated tomato plants (Figure 3). Furthermore, three or six sprays of all the tested treatments increased the activity of the chitinase enzyme at two and five days post inoculation with *A. alternata*. Six sprays of salicylic acid resulted in the maximum increase in chitinase activity in inoculated tomato fruits, at five days of storage, compared with the other tested treatments. Additionally, spraying tomato plants three or six times with chitosan increased the chitinase activity in inoculated tomato fruits, at two and five days of storage, compared to the control.

In the case of lithovit and potassium thiosulfate, six sprays were required to show an increase in the chitinase activities in tomato fruits at two and five days post inoculation. The least activity of chitinase was recorded with three sprays of potassium thiosulfate at two days of storage. For naturally infected tomato fruits, both three and six sprays of all the tested chemicals increased the activity of chitinase in tomato fruits at two and five days of storage, compared to the control treatment. The highest activity of chitinase enzyme in naturally infected tomato fruits was recorded with six sprays of chitosan at five days of storage.

On the other hand, the least activity of chitinase was recorded with three sprays of lithovit at two days of storage. In general, data revealed that the activity of chitinase was higher in control treatment under artificial infection than natural infection. The obtained results could be interpreted in light of the findings of [65], who reported that salicylic acid caused the accumulation of chitinase in tomato plants, which conferred resistance against *Phytophthora infestans*. On other hand, the application of SA increased the activity of PR-proteins such as chitinase and β 1,3-glucanase, which inhibited the process of pathogenesis in susceptible tomato cultivars to exhibit the resistance [66] These results also conform with the findings of [67], who observed that the treatment of grapevine leaves with chitosan resulted in a marked increase in chitinase activities, which in turn might improve the resistance to grey mold. On the other hand, chitosan treatment can induce plant defense through the stimulation of enzymes related to pathogenesis and prolong the shelf life of fruits and vegetables [68].

Regarding catalase activity, all the four tested chemicals positively influenced the activities of the enzyme in fruits of treated tomato plants (Figure 4). In this respect, spraying tomato plants three times with salicylic acid, lithovit, and chitosan showed a marginal increase in the catalase activity in tomato fruits at two days post-inoculation with *A. alternata*, compared to the control. After six sprays, the same treatments increased catalase activity in treated tomato fruits at two days post inoculation with the same pathogen. The maximum increase in the catalase activity in tomato fruits was observed in plants sprayed six times with salicylic acid and at five days of storage post inoculation. On the other hand, catalase activities decreased in tomato fruits treated with three sprays of all the tested chemicals at five days post inoculation with the pathogen, compared to the control. Compared to lithovet and potassium thiosulfate, six sprays of chitosan increased the catalase activity in tomato fruits inoculated with *A. alternata* at two and five days of storage. In the case of naturally infected tomato fruits, both three- and six-spray treatments of all the tested chemicals increased the activity of catalase at both two and five days of storage, except for three sprays of lithovet at five days post inoculation. The highest increase in catalase activity of tomato fruits was recorded with six sprays of chitosan at five days, followed by potassium thiosulfate. These results are in harmony with those recorded by [69], who reported that catalase (CAT) activity was increased in peach fruits treated with salicylic acid. Additionally, Xu and Tian [70] reported that treatment with salicylic acid increased the CAT activity and stimulated the expression of *CAT* genes in sweet cherry fruits. On the other hand, pre-harvest spraying of rubber trees with salicylic acid induced the accumulation of catalase and peroxidase in the leaves [71]. Additionally, Hortensia et al. [72] observed that the exogenous application of chemical elicitors such as chitosan and salicylic acid during different stages in the development of tomato fruits significantly increased the activity of catalase and peroxidase enzymes in the fruit tissue. On the other hand, chitosan treatment plays an important role in resistance to oxidation by improving CAT activity in strawberry fruits during cold storage [73]. Cucumber coated with 1.5% chitosan demonstrated high levels of activity of ascorbate peroxidase (APX) and catalase (CAT) [74].

### 3.6. Effect of the Pre-Harvest Foliar Application of the Tested Chemicals on mRNA Expression of Defense-Related Genes in Treated Tomato Plants

Quantitative analysis of mRNA levels of the studied defense-associated genes confirmed that the plant SAR was regulated by the levels of the systemic signal molecules, potassium thiosulfate, lithovit, salicylic acid, and chitosan. Seven tomato defense-associated genes—*RAP*, *XET–2*, *ACS–2*, *PINII*, *PAL5*, *LOXD*, and *PR1*—were differently expressed under elicitation by the four studied elicitors, these genes exhibited long lasting transcript accumulation. The targeted genes belonging to the ethylene pathway (*RAP*, *XET–2*, and *ACS–2*) were upregulated in tomato plants sprayed three or six times with either chitosan, lithovit, salicylic acid, or potassium thiosulfate at their respective concentrations (Figure 5A–C, respectively). The highest levels of mRNA transcripts for *RAP*, *XET–2*, and *ACS–2* genes were 16.2-, 27.3-, and 25.8-fold higher, respectively, in response to the pre-harvest treatment with six sprays of 4 g/L chitosan under field conditions. In general, higher expression signals of these genes were observed in tomato plants treated with six sprays of all the tested chemical elicitors, compared to three sprays. Treatment with 4 g/L chitosan was more potent in inducing the expression of tomato resistance genes.

Our findings of the differential expression of the targeted defense genes agreed with those of Herman et al. [75] who reported that SA stimulated the production of plant defense-related compounds and improved SAR. They reported that SAR was distinct from other plant defense responses by the indigenous and systemic stimulation of specific pathogenesis-related genes (*PR* genes). Furthermore, the stimulation of the *RAP* gene observed in this study agreed with the work of [76] who demonstrated that the ethylene response factor gene (*RAP*) was an integral constituent of these signaling cascades as it regulated the expression of a wide variety of genes related to stress response and development through different mechanisms. In addition, the induction of the *XET–2* gene in this study agreed with the findings of [77], who suggested that the *XET–2* gene encoded xyloglucan endotransglycosylase. This gene also stimulates the xyloglucan endo-cleavage polymers as well as consequent transfers of the newly generated reducing ends to other polymeric or oligomeric xyloglucan molecules. Thus, *XET* protein action is a potential tool to achieve controlled wall relaxation during turgor-driven expansion by rearranging the load-bearing xyloglucan cross-links between cellulose microfibrils. Moreover, it may stimulate the molecular reactions essential to integrate budding xyloglucan polysaccharides into the existing cell wall, thus maintaining the width and integrity of the cell wall. On the other hand, regulation of the *ACS–2* gene in this study agreed with the findings of [78], who reported that *ACS–2* was an ethylene biosynthesis gene. A developmental alteration from ethylene system I to ethylene system II occurred in tomato fruits during ripening. The system I is a process of ethylene auto-inhibitory, while system II is a process of ethylene auto-catalytic. Several ethylene-related genes such as *ACS2* and *ASC4* can be up-regulated by the transition from system I to system II. Moreover, Alexandersson et al. [79] reported that the expression of pathogenesis-related genes involved in the defense mechanism of tomato fruits have significantly enhanced the resistance against various pathogens.

In addition, the expression of *PINII*, *PAL5*, *LOXD*, and *PR1* genes related to the jasmonate and salicylate pathways were also studied in response to the four tested chemical inducers for SAR in tomato plants. The expressions of these genes were upregulated by treatment with either three or six sprays with either chitosan, lithovit, salicylic acid, and potassium thiosulfate at their respective concentrations (Figure 6A–D, respectively).

In this respect, the highest level of gene expression of *PINII*, *PAL5*, *LOXD*, and *PR1* resistance genes (19.6-, 15.2-, 20.2-, and 18.3-fold increase, respectively) was observed in tomato plant tissues treated with six sprays of chitosan. From the results, the induction of defense-related genes could partially explain the enhanced control of tomato black mold disease, thus enhancing the productivity, shelf life, and quality of fruits. The differential expression of the *PINII* gene observed in this study agreed with the findings of Turra and Lorito, [80] who investigated the expression of the protease inhibitor 1 family the *PINII* gene, in response to various environmental factors and injury. However, activation of the *PINII* gene is also induced by bacterial and viral infections, and insect and nematode attacks, as well as the application of fungal elicitors. Furthermore, several chemical and physical stimuli, such as sucrose, oligogalacturonides, systemin, abscisic acid, auxin, ethylene, jasmonic acid, heat, and electrical currents might upregulate the transcription of *PINII*. On the other hand, aspirin and SA can repress injury-, jasmonic acid-, or systemin-mediated *PINII* gene stimulation. Additionally, the stimulation of the gene *PAL5* observed in this study was in line with the findings of Chandrasekaran and Chun, [81] who reported that phenyl ammonia lyase (PAL) is a key enzyme involved in phenylpropanoid metabolism leading to the production of defensive compounds (lignins, coumarins, flavonoids, and phytoalexins). PAL is a vital protein for plant growth as well as defense against pathogens. The induced systemic resistance increases the pathogenesis-related proteins (PR), such as PR–1 protein. Furthermore, upregulation of the *LOXD* gene observed in this study agreed with the findings of Safaie-Farahani and Taghavi [82], who reported that lipoxygenases (LOXs) may act as signaling molecules involved in structural and metabolic changes in plants, resulting in resistance against the pathogen. The activation of LOX in plants in response to environmental and biotic stresses has been reported [83]. The results of Yan et al. [83] indicated that TomLoxD was involved in wound-induced JA biosynthesis, and highlighted the application potential of this gene for crop protection against insects and pathogens. The researchers also reported that the overexpression of TomLoxD led to elevated wound-induced JA biosynthesis, increased expression of wound-responsive genes, and therefore, enhanced resistance to insect herbivory attack and necrotrophic pathogen infection.

## 4. Conclusions

Based on in vitro studies, salicylic acid was the most effective among all treatments, as it completely inhibited the growth of *A. alternata* at all concentrations. Results of the in vivo studies showed that six applications of all plant resistant inducers achieved better inhibition of *A. alternata* infection, compared to three applications. Foliar application with either chitosan, salicylic acid, lithovit, or potassium thiosulfate gave the highest vegetative growth, total yield, best quality of tomato fruits, and promoted the expression of defense-associated genes, especially when sprayed six times rather than three times, compared to the control (distilled water). Chitosan followed by salicylic acid was the most effective among the tested elicitors in controlling the black mold rot in tomato fruits caused by *A. alternata*. The maximum increase in chitinase and catalase activity of tomato fruits was observed at five days after inoculation, following treatment with six sprays of salicylic acid. In addition, chitosan treatment yielded the highest acquired systematic resistance, compared to salicylic acid, lithovit, and potassium thiosulfate treatments. This resistance was accompanied by the augmented expression of seven defense-associated genes (*RAP*, *XET–2*, *ACS–2PINII*, *PAL5*, *LOXD*, and *PR1*) probably showing the induction of SAR. Chitosan followed by salicylic acid was the most effective among the tested elicitors in controlling the black mold rot in tomato fruits. This study recommends pre-harvest chitosan and salicylic acid field application with six sprays as effective safe alternatives to fungicides against black mold disease to boost sustainable productivity of tomato fruits.

## Figures and Tables

**Figure 1 plants-09-00808-f001:**
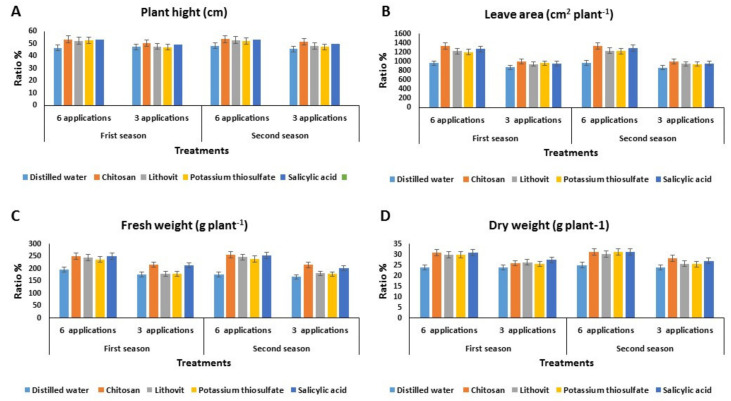
The effect of the four tested chemicals as pre-harvest foliar application on vegetative growth characters of tomato plants during two growing seasons 2017 and 2018. (**A**) Plant height (cm), (**B**) leaf area (cm^2^ plant^−1^), (**C**) fresh weight (g plant^−1^), and (**D**) dry weight (g plant^−1^).

**Figure 2 plants-09-00808-f002:**
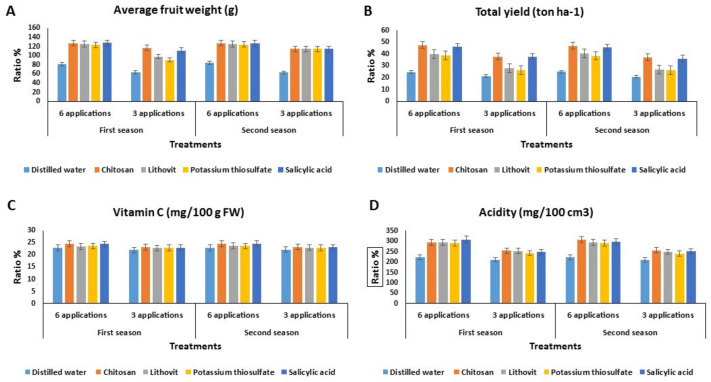
The effect of the four tested chemicals as pre-harvest foliar application on yield and quality of tomato fruits during two growing seasons 2017 and 2018. (**A**) Average fruit weight (g), (**B**) total yield (ton ha^−1^), (**C**) vitamin C (mg/100 g FW), and (**D**) acidity (mg/100 cm^3^).

**Figure 3 plants-09-00808-f003:**
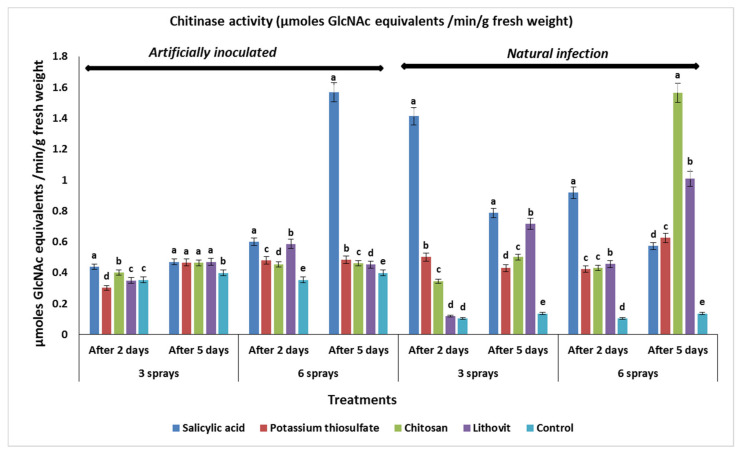
The effect of the tested chemicals as pre-harvest foliar application at 3 and 6 sprays on chitinase activity in the treated tomato fruits, naturally and artificially infected with *A. alternata*, at different periods of incubation (after two or five days) (as µmoles GlcNAc equivalents/min/g fresh weight). Within each incubation period (after two or five days), the same letters indicate no significant difference among the treatments at (*p* < 0.05).

**Figure 4 plants-09-00808-f004:**
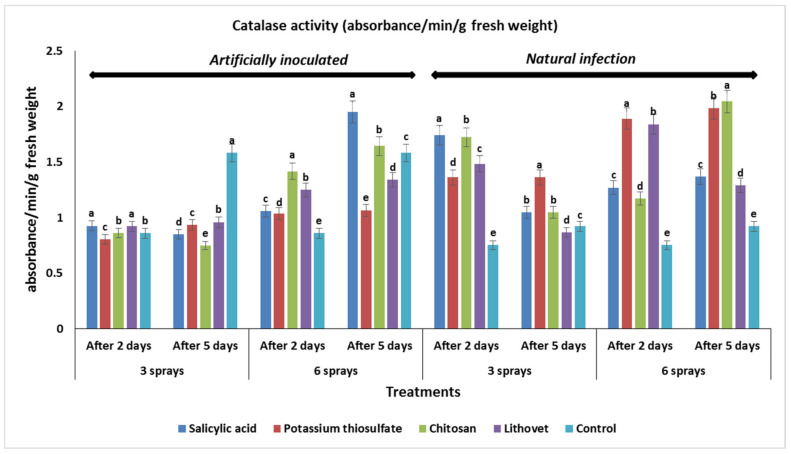
The effect of tested chemicals as pre-harvest foliar application at 3 and 6 sprays on catalase activity in the treated tomato fruits, naturally and artificially infected with *A. alternata*, after different periods of incubation (absorbance/min/g fresh weight). Within each incubation period (after two or five days), the same letters indicate no significant difference among treatments at (*p* < 0.05).

**Figure 5 plants-09-00808-f005:**
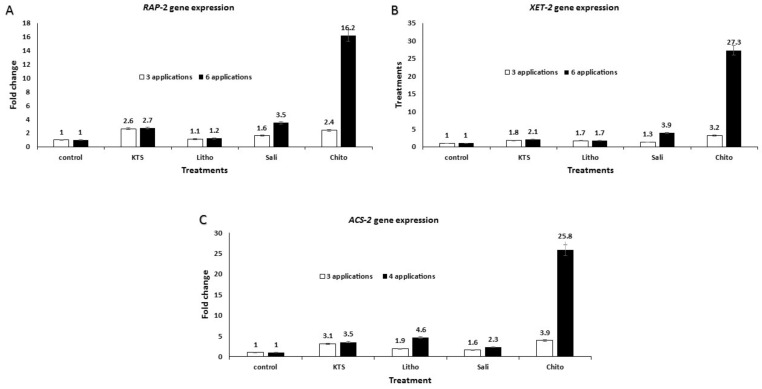
The effect of the tested chemicals as a pre-harvest foliar application on gene expression profiling of associated-resistance genes in treated tomato plants using qRT-PCR analysis. (**A**) *RAP* gene, (**B**) *XET–2* gene, and (**C**) the *ACS–2* gene fold change after three- and six-spray applications with K-thiosulfate (KTS)/4 g L^−1^, lithovit (Litho)/4 g L^−1^, salicylic acid (Sali)/1.5 g L^−1^ and chitosan (Chito)/4 g L^−1^, respectively. The *Actin* gene was used as an internal reference gene for data normalization [24].

**Figure 6 plants-09-00808-f006:**
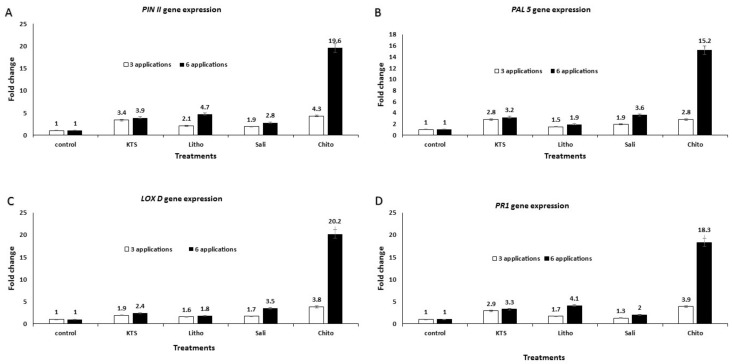
The effect of the tested chemicals as a pre-harvest foliar application on gene expression profiling of associated-resistance genes in the treated tomato plants using qRT-PCR analysis. (**A**) *PINII* gene, (**B**) *PAL5* gene, (**C**) *LOXD* gene and (**D**) *PR1* gene fold change after three- and six-spray applications with K-thiosulfate (KTS)/4 g L^−1^, Lithovit (Litho)/4 g L^−1^, salicylic acid (Sali)/1.5 g L^−1^, and chitosan (Chito)/4 g L^−1^, respectively. The *Actin* gene was used as an internal reference gene for data normalization [24].

**Table 1 plants-09-00808-t001:** Physical and chemical properties of the used soil for transplanting tomato seedlings (*Lycopersicon esculentum* Mill cv. Super Strain B).

Soil Texture				Soil Parameters				Soil Available Macronutrients (mg kg^−1^)		
Sand%	Silt%	Clay%	Texture *	PH	EC/dSm^−1^	O.M/g kg^−1^	CaCO/g kg^−1^	N	K	P
24.4	24.6	51	Clay loam	7.9	2.16	1.41	1.53	22.5	9.1	120

* Texture according to the international soil texture Triangle (Moeys 2016); EC of paste extract; Extractants are KCl (K), NaHCO_3_ (P), NH_4_Ac (N).

**Table 2 plants-09-00808-t002:** Oligonucleotides with sequences used for quantitative real time polymerase chain reaction (qRT-PCR).

Gene Name	Primer Forward	Primer Reverse	Reference
*RAP*	5′-aaagaaccatctgtggcgtgtgag–3′	5′-cgaatcttgtaagcggcttggtca–3′	Upadhyay et al. (2014) [24]
*XET–2*	5′-tggaggagattctgctggtgttgt–3′	5′-tctgtctcctttgcctcctgtgaa–3′	Upadhyay et al. (2014) [24]
*ACS–2*	5′-ttccatcactgcagctttgcttcg–3′	5′-tttgtttgggccagcttctctctc–3′	Upadhyay et al. (2014) [24]
*PAL5*	5′-gacagcaggaaggaatccaa–3′	5′-caaccaaatagggattcgaca–3′	Tolba et al. (2018) [25]
*LOXD*	5′-ttggcaccaagttcaggccc–3’	5′-tggacttaagctagtattag–3′	Tolba et al. (2018) [25]
*PR1*	5′-tgccaagaccggtggtaatttc–3′	5′-tgcccgctagcacattggt–3′	Tolba et al. (2018) [25]
*PINII*	5′-ttgttgtgcaggcagtaagg–3′	5′-ttcaataattacgcgtgagcc–3′	Tolba et al. (2018) [25]
*Actin* (reference gene)	5′-ttgccgcatgccattct–3′	5′-tcggtgaggatattcatcaggtt–3′	Upadhyay et al. (2014) [24]

**Table 3 plants-09-00808-t003:** Effect of tested chemicals on the in vitro growth of *Alternaria alternata.*

Tested Chemicals	Conc. mg/mL	Linear Growth	
		mm	Efficacy%
**Salicylic acid**	0.75	0.0 ^f^	100.0
	1.5	0.0 ^f^	100.0
**Potassium thiosulfate**	2	55.2 ^cd^	38.7
	4	45.7 ^e^	49.3
**Chitosan**	2	57.3 ^c^	36.3
	4	50.7 ^de^	43.7
**Lithovit**	2	68.7 ^b^	23.7
	4	56.3 ^cd^	37.4
**Control**		90.00 ^a^	

Within each column, the same letters indicate no significant difference among the treatments at (*p* < 0.05).

**Table 4 plants-09-00808-t004:** The effect of tested chemicals as pre-harvest foliar applications on tomato black rot during storage under cold conditions during season 2018.

Tested Chemicals	Six Application							
	*Alternaria alternata*				Natural Infection			
	7 Days from Storage		15 Days from Storage		7 Days from Storage		15 Days from Storage	
	D.S	EF%	D.S	EF%	D.S	EF%	D.S	EF%
**Salicylic Acid**	0.00 ^b^	100.00	5.51 ^ef^	91.91	0.00b	100.00	0.00 ^c^	100.00
**Potassium Thiosulfate**	1.33 ^b^	86.20	9.33 ^cde^	86.32	0.00 ^b^	100.00	1.03 ^bc^	95.99
**Chitosan**	0.00 ^b^	100.00	4.12 ^f^	93.96	0.00 ^b^	100.00	0.00 ^c^	100.00
**Lithovit**	1.24 ^b^	87.14	6.95 ^def^	89.81	0.00 ^b^	100.00	0.00 ^c^	100.00
**Three Applications**								
**Salicylic Acid**	0.00 ^b^	100.00	10.42 ^cd^	84.72	0.00 ^b^	100.00	2.08 ^bc^	91.89
**Potassium Thiosulfate**	1.77 ^b^	81.64	13.53 ^c^	80.16	0.00 ^b^	100.00	3.80 ^bc^	85.19
**Chitosan**	1.03 ^b^	89.32	7.64 ^def^	88.80	0.00 ^b^	100.00	2.73 ^bc^	89.36
**Lithovit**	1.65 ^b^	82.88	17.89 ^b^	73.76	0.00 ^b^	100.00	4.50 ^b^	82.46
**Control**	9.64 ^a^		68.19 ^a^		2.05 ^a^		25.66 ^a^	

Within each column (for six and three applications), the same letters indicate no significant difference among treatments at (*p* < 0.05). EF%: efficacy as a percentage. D.S: disease severity. Control was infected with *A. alternata* or without infection and un-treated.

**Table 5 plants-09-00808-t005:** The effect of tested chemicals as pre-harvest foliar applications on tomato black rot during storage under cold conditions during season 2019.

Tested Chemicals	Six Application							
	*Alternaria alternata*				Natural Infection			
	7 Days from Storage		15 Days from Storage		7 Days from Storage		15 Days from Storage	
	D.S	EF%	D.S	EF%	D.S	EF%	D.S	EF%
**Salicylic Acid**	0.00 ^b^	100.00	6.19 ^cd^	90.52	0.00 ^a^	100.00	0.00 ^b^	100.00
**Potassium Thiosulfate**	1.53 ^b^	85.99	10.63 ^bcd^	83.72	0.00 ^a^	100.00	1.39 ^b^	94.76
**Chitosan**	0.00 ^b^	100.00	3.81 ^d^	94.17	0.00 ^a^	100.00	0.00	100.00
**Lithovit**	1.39 ^b^	87.27	7.25 ^cd^	88.90	0.00 ^a^	100.00	0.00	100.00
**Three Application**								
**Salicylic Acid**	0.00 ^b^	100.00	9.20 ^bcd^	85.91	0.00 ^a^	100.00	1.93 ^b^	92.73
**Potassium Thiosulfate**	1.81 ^b^	83.42	12.29 ^bc^	81.18	0.00 ^a^	100.00	3.33 ^b^	87.45
**Chitosan**	1.16 ^b^	89.38	7.16 ^cd^	89.04	0.00 ^a^	100.00	2.42 ^b^	90.88
**Lithovit**	2.58 ^b^	80.95	15.03 ^b^	76.99	0.00 ^a^	100.00	4.52 ^b^	82.96
**Control**	10.92 ^a^		65.31 ^a^		2.08 ^a^		26.53 ^a^	

Within each column (for six and three applications), the same letters indicate no significant difference among treatments at (*p* < 0.05). EF%: efficacy as a percentage. D.S: disease severity. Control was infected with *A. alternata* or without infection and un-treated.

## Data Availability

All data generated or analyzed during this study are included in this published article.

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
