# Peer review of "Comparative Analyses of Four Chemicals Used to Control Black Mold Disease in Tomato and Its Effects on Defense Signaling Pathways, Productivity and Quality Traits"

_plants, 2020, doi:10.3390/plants9070808_

Round 1

Reviewer 1 Report

The paper investigates the effectiveness of chitosan, salicylic acid, lithovit and potassium thiosulfate used to control tomato Black mold disease and enhance productivity and quality, as well as improving the induced resistance against post-harvest infection caused by Alternaria alternata.

The manuscript is well written, the experiments were carried out fine with good analysis and has merits for publication in Plants. However, the document still needs some minor corrections and clarifications.

-Please, explain why three and six spray applications with chitosan, salicylic acid, lithovit, and K-thiosulfate were used?. Which was the criteria four using it?.

-Lines 515-517: The authors describe that a positive correlation was observed between the molecular performance and plant behavior in controlling tomato black mold disease. The authors could include a correlation analysis to confirm this conclusion.

-Please, write scientific names of species in cursive in the manuscript, In vivo, in vitro.

Author Response

Comments and replies

We would like to thank the reviewers for the valuable comments and constructive suggestions to our manuscript, which greatly helped to enhance the quality of our manuscript.

For the provided specific points:

we have mad the requested modifection in yellow highlight and hope the new version would meet the requirements.

Comments and replies

We would like to thank the reviewers for the valuable comments and constructive suggestions to our manuscript, which greatly helped to enhance the quality of our manuscript.

For the provided specific points:

we have mad the requested modifection in yellow highlight and hope the new version would meet the requirements.

Reviewer comments

Authors replies

Reviewer 1 comments:

-Please, explain why three and six spray applications with chitosan, salicylic acid, lithovit, and K-thiosulfate were used?. Which was the criteria four using it?.

As a new treatments to investigate potentiate the benifits from the used chemical by applying it three and six times. Where, the three time spry start on the folowering stage of the plant for inducing the SAR, while six time sprys begin in early developmental stage of the plant (21 days after planting) to stimulte plant growth as well as SAR stimulation

2- Lines 515-517: The authors describe that a positive correlation was observed between the molecular performance and plant behavior in controlling tomato black mold disease. The authors could include a correlation analysis to confirm this conclusion.

- we replaced " positive correlation" by " positive relation"

3, Please, write scientific names of species in cursive in the manuscript, In vivoin vitro.

-       It has been corrected

Reviewer comments

Authors replies

Reviewer 1 comments:

-Please, explain why three and six spray applications with chitosan, salicylic acid, lithovit, and K-thiosulfate were used?. Which was the criteria four using it?.

As a new treatments to investigate potentiate the benifits from the used chemical by applying it three and six times. Where, the three time spry start on the folowering stage of the plant for inducing the SAR, while six time sprys begin in early developmental stage of the plant (21 days after planting) to stimulte plant growth as well as SAR stimulation

2- Lines 515-517: The authors describe that a positive correlation was observed between the molecular performance and plant behavior in controlling tomato black mold disease. The authors could include a correlation analysis to confirm this conclusion.

- we replaced " positive correlation" by " positive relation"

3, Please, write scientific names of species in cursive in the manuscript, In vivoin vitro.

-       It has been corrected

Reviewer 2 Report

The work from Moustafa et al. evaluates the effect of the application of diverse chemicals on tomato growth and quality, shelf life, and as inducers of systemic acquired resistance. Also, the authors evaluate their function as in vitro inhibitors of fungal growth. The work is mostly well written and the experiments have relatively good technical standards, but the work itself is a mixture of many experiments from which different conclusions can be obtained if they are analyzed separately and to a deeper extent. 

The title of the manuscript refers to gene expression signals (which should be replaced as gene expression levels or something similar) which is a very small part of the manuscript. Also, the chemicals are referred to as systemic acquired resistance inducers, but as described in the manuscript they also exert various other roles in plant growth.

The results have mostly agronomical implications, and very little is addressed about the potential mechanisms. Authors should explicitly state that they are evaluating the chemicals in different contexts and probably exerting different roles in each of the experiments. For example:

1) In vitro inhibition of fungal growth is directly affecting the fungus and differs from priming plants for acquired resistance.

2) If chemicals affect growth parameters and fruit quality so dramatically, how can you differentiate their effect as plant growth regulators from systemic resistance inducers?

I think authors should think about which are the questions trying to be addressed, which of these questions are answered from each of the experiments, and try to change the structure accordingly in order to highlight the main findings and implications.

Other concerns:

Please consider changing the title of the manuscript in order to accurately describe the findings and not especially highlight the gene expression results.

In the abstract, please add some more context about the chemicals and make a better effort to summarize the findings.

In the introduction, please add references to inducers used to control Alternaria alternate (lines 51-52).

In the qPCR analysis, how did you choose Actin as reference gene? Is the efficiency of the reactions similar?

Author Response

We would like to thank the reviewers for the valuable comments and constructive suggestions to our manuscript, which greatly helped to enhance the quality of our manuscript.

For the provided specific points:

we have mad the requested modifection in yellow highlight and hope the new version would meet the requirements.

Reviewer comments

Authors replies

Reviewer 2 comments:

The title of the manuscript refers to gene expression signals (which should be replaced as gene expression levels or something similar)

Modified title

Molecular Signaling of Some Safe Inducers Used to Control Tomato Black Mold Disease, and Enhance Productivity and Quality

(already added to the MS revised copy)

- Authors should explicitly state that they are evaluating the chemicals in different contexts and probably exerting different roles in each of the experiments. For example:

1) In vitro inhibition of fungal growth is directly affecting the fungus and differs from priming plants for acquired resistance.

2) If chemicals affect growth parameters and fruit quality so dramatically, how can you differentiate their effect as plant growth regulators from systemic resistance inducers?

-       We used the chemicals fo both purpose and we explored it. Also, we used six time spray for promiting the growth of the plant as well post harvest resistance induction (we explaned this in the results and discussion section)

-In the abstract, please add some more context about the chemicals and make a better effort to summarize the findings.

It has been corrected

-In the introduction, please add references to inducers used to control Alternaria alternate (lines 51-52).

We merged the sentence within the next paragrph where it belongs to it

- In the qPCR analysis, how did you choose Actin as reference gene? Is the efficiency of the reactions similar?

Actin (a housekeeping gene) was used as reference gene (endogenous control) in this study since its expression remained more stable during the experiments

Reviewer 3 Report

Having read the manuscript I commend the authors for a thorough and extensive study.  My criticisms are limited to my understanding of the results as presented.  Most importantly, in Tables 5 and 6 the results after 7 and 15 days in storage do not support any claim that chitosan is more efficaceous than the other materials.  Although chitosan is more consistent the statistical outcome suggests no significant difference with six applications.  I felt that these results would benefit from a more clear presentation ... perhaps graphical?  The results do point towards the conclusion stated in the abstract but agree more with the conclusions at the end of the manuscript.  It was also very interesting to see the contrast in chitinase and catalase levels between artificial and natural infection...I was disappointed that the authors did not explain this further. 

The presentation of the results was confusing.   In Tables 4 and 5 results are compared among 2 columns including 3 and 6 applications followed by  1 column for the averages.  I feel the authors could present this data in a more understandable fashion (I suggest the use of boxes within the table...see attached file).  Tables 6a and b could be combined and the EF% column deleted since no stats are conducted.  Overall, the data was time consuming to interpret and the presentation could make this much easier.

I have included several comments on the attached manuscript.

I am recommending major revision only because the formatting and some of the interpretations are problematic.  The manuscript presents valuable, well executed experiments with valuable data.

Author Response

We would like to thank the reviewers for the valuable comments and constructive suggestions to our manuscript, which greatly helped to enhance the quality of our manuscript.

For the provided specific points:

we have mad the requested modifection in yellow highlight and hope the new version would meet the requirements.

Reviewer comments

Authors replies

Reviewer 3 comments:

- Most importantly, in Tables 5 and 6 the results after 7 and 15 days in storage do not support any claim that chitosan is more efficacious than the other materials. 

Table 5 does not include 7 and 15 dys storage treatments, results in table 6a and 6b, explored that chitosan was more efficacious after 15 days storage than other treatments followed by salicylic acid in three and six applications specially under artificial infection. (all these are included in the results and discusion section, in addtion the effect of all tested treatments)

- Although chitosan is more consistent the statistical outcome suggests no significant difference with six applications. 

In general, among all tested treatments there were no statistical differences specially after seven days cold storage. (already added to the results and discusion section)

- The results do point towards the conclusion stated in the abstract but agree more with the conclusions at the end of the manuscript. 

It has been corrected

- The presentation of the results was confusing.   In Tables 4 and 5 results are compared among 2 columns including 3 and 6 applications followed by  1 column for the averages.  I feel the authors could present this data in a more understandable fashion (I suggest the use of boxes within the table...see attached file). 

The requested modifications has been done

The revised tables were supplied to the the revised manuscript.

- Tables 6a and b could be combined and the EF% column deleted since no stats are conducted.  Overall, the data was time consuming to interpret and the presentation could make this much easier.

We renamed bothe tables to be table 6 and table 7.

The EF% is an important parametr for the tested chemicals in vitro and in vivo

I have included several comments on the attached manuscript.

We applied all the requested comments and changes

Round 2

Reviewer 2 Report

Please rephrase the title, especially don’t use the word “some” as it is too vague. I suggest:

Comparative analyses of four chemicals used to control Black Mold Disease in tomato and its effects on defense signaling pathways, productivity and quality traits

Abstract

Please introduce the chemicals and their use before start summarizing the results. Please reduce the results showed in the abstract to highlight only the most relevant ones.

Line 45: please rephrase with: “However, production is threatened by many different pathogens that affect quality and productivity.” Or similar.

Line 60: Please remove “on the other hand”.

Line 65: Please rephrase “weight gain” with “biomass accumulation” or similar.

Line 67: Please add “Lithovit is a foliar fertilizer that improves plant growth…”

Line 73: Please rephrase with “It has been proposed that

Lines 74-79. Please replace these examples by “Potassium (K+) is a macronutrient required for fundamental physiological and molecular processes in plants.” Or similar.

Lines 82-84. Please introduce SAR before talking about the genes. Something like: “Systemic acquired resistance (SAR) is a mechanism by which plant defenses are preconditioned by a prior treatment that results in enhanced resistance against subsequent pathogen infections. Although the molecular details of the signaling machinery are poorly understood, previous studies have shown…” or similar.

Lines 90-96. Please rephrase, you are studying the chemicals in many different contexts. Something like “This study aimed to analyze the effectiveness of chitosan, salicylic acid, lithovit, and potassium thiosulfate in improving tomato productivity and quality. We assessed if these chemicals affect fungal growth, plant growth parameters, cold storage period, yield components, and fruit quality, as well as modifying post-harvest resistance to Alternaria alternata infection. Finally, we addressed whether we could identify the induction of SAR-related defense genes by analyzing the expression of RAP, XET–2, ACS–2, PINII, PAL5, LOXD, and PR1” or similar.

Line 132. Please rephrase, as the chemicals are not acting as inducers in this assay. “The effect of the chemicals on the growth of Alternaria alternata…” or similar.

Line 245. Please add a suitable reference showing that Actin is a good reference gene in similar conditions as the ones assayed in this work.

Line 288. Please remove “the”.

Line 310. A dot is missing after “[48]”.

Lines 312-313. It is not clear what you want to say about clay soil, please rephrase. 

Line 4. Please replace “evident from” with “observed in”.

Line 5. Please replace “outcome” with “fruit production”

Tables 4 and 5. I strongly suggest to the authors to replace these tables with graphical charts. This will facilitate the analysis and will allow incorporating the dispersion of the data (standard error/standard deviation), which is not incorporated in the tables. I also suggest using a graphical representation that includes individual data points as proposed in this work https://journals.plos.org/plosbiology/article?id=10.1371/journal.pbio.1002128.

Tables 6 and 7. Please try to join the data in one single table summarizing the results from both seasons. If possible, also use a graphical representation.

Line 21. Please include a brief introduction about why are you studying chitinase and catalase activities.

Figures 1 and 2. Please replace the left panel named “Alternaria alternata” with “artificially inoculated” or something similar. Please also consider including the individual data points in the graphs as suggested before.

Line 83-84. Please replace “Gene Expression Signals of Resistance Genes in Treated Tomato Plants” with “mRNA expression of defense-related genes” or something similar.

Figures 3 and 4. Please include the relative expression of the controls and indicate which of the treatments actually present statistical differences from the control treatment. Please replace the “3 days and 6 days” labels with “3 spray applications and 6 spray applications” if this is the case.

Line 126. Please remove “differential”.

Lines 139-140. Please rephrase “positive relation was observed between the molecular performance and plant behavior in controlling tomato black mold disease” with “the induction of defense-related genes could partially explain the enhanced control of tomato black mold disease” or similar.

Lines 177-178. Please replace “This resistance was induced through the expression of the seven target defense genes” by “This resistance was accompanied by the augmented expression of seven defense-associated genes (RAP, XET–2, ACS–2PINII, PAL5, LOXD, and PR1) probably showing the induction of SAR” or similar.

Lines 179-183. The last two sentences are almost the same, please rephrase. On the other hand, I think any producer will select the chemical based on gene expression profiles, so I suggest selecting the best inducer based on overall productivity and quality.

Author Response

"Authors’ response to reviewers’ comments"

Gene Expression Signals of Some Safe Inducers Used to Control Tomato Black Mold Disease, and Enhance Productivity and Quality

Hoda A. S. El-Garhy1, Fayz A. Abdel-Rahman2, Abdelhakeem S. Shams3,
Gamal Osman 4,5,6,* and Mahmoud M. A. Moustafa1

Comments and replies

We would like to thank the reviewer 2 for the valuable comments and constructive suggestions to our manuscript, which greatly helped to enhance the quality of our manuscript.

For the provided specific points:

we have mad the requested modifection and hope the revised version would meet the requirements.

 Reviewer 2 comments:

Authors replies

1- Please rephrase the title, especially don’t use the word “some” as it is too vague. I suggest:

Comparative analyses of four chemicals used to control Black Mold Disease in tomato and its effects on defense signaling pathways, productivity and quality traits

We accepted the suggested title and repaced the old one by it, thank you for it

2- Abstract

Please introduce the chemicals and their use before start summarizing the results. Please reduce the results showed in the abstract to highlight only the most relevant ones.

- done

3- Line 45: please rephrase with: “However, production is threatened by many different pathogens that affect quality and productivity.” Or similar.

- done

4- Line 60: Please remove “on the other hand”.

done

5- Line 65: Please rephrase “weight gain” with “biomass accumulation” or similar.

-       Done  

6- Line 67: Please add “Lithovit is a foliar fertilizer that improves plant growth

- done

7- Line 73: Please rephrase with “It has been proposed that

done

8- Lines 74-79. Please replace these examples by “Potassium (K+) is a macronutrient required for fundamental physiological and molecular processes in plants.” Or similar.

done

9- Lines 82-84. Please introduce SAR before talking about the genes. Something like: “Systemic acquired resistance (SAR) is a mechanism by which plant defenses are preconditioned by a prior treatment that results in enhanced resistance against subsequent pathogen infections. Although the molecular details of the signaling machinery are poorly understood, previous studies have shown…” or similar.

done

10- Lines 90-96. Please rephrase, you are studying the chemicals in many different contexts. Something like “This study aimed to analyze the effectiveness of chitosan, salicylic acid, lithovit, and potassium thiosulfate in improving tomato productivity and quality. We assessed if these chemicals affect fungal growth, plant growth parameters, cold storage period, yield components, and fruit quality, as well as modifying post-harvest resistance to Alternaria alternata infection. Finally, we addressed whether we could identify the induction of SAR-related defense genes by analyzing the expression of RAP, XET–2, ACS–2, PINII, PAL5, LOXD, and PR1” or similar.

done

11- Line 132. Please rephrase, as the chemicals are not acting as inducers in this assay. “The effect of the chemicals on the growth of Alternaria alternata…” or similar.

done

12- Line 245. Please add a suitable reference showing that Actin is a good reference gene in similar conditions as the ones assayed in this work.

Done, the requested reference (Upadhyay et al. 2014) was added to M&M section and to table 2

13- Line 288. Please remove “the”

done

14- Line 310. A dot is missing after “[48]”.

done

15- Lines 312-313. It is not clear what you want to say about clay soil, please rephrase. 

rephrased

16- Line 4. Please replace “evident from” with “observed in”.

done

17- Line 5. Please replace “outcome” with “fruit production”

done

18- Tables 4 and 5. I strongly suggest to the authors to replace these tables with graphical charts. This will facilitate the analysis and will allow incorporating the dispersion of the data (standard error/standard deviation), which is not incorporated in the tables. I also suggest using a graphical representation that includes individual data points as proposed in this work https://journals.plos.org/plosbiology/article?id=10.1371/journal.pbio.1002128.

done

19- Tables 6 and 7. Please try to join the data in one single table summarizing the results from both seasons. If possible, also use a graphical representation.

If we joined them the resulted table will apear more coplex, so, it will be better to keep both tables separated

20- Line 21. Please include a brief introduction about why are you studying chitinase and catalase activities.

done

21- Figures 1 and 2. Please replace the left panel named “Alternaria alternata” with “artificially inoculated” or something similar. Please also consider including the individual data points in the graphs as suggested before.

done

22- Line 83-84. Please replace “Gene Expression Signals of Resistance Genes in Treated Tomato Plants” with “mRNA expression of defense-related genes” or something similar.

done

23- Figures 3 and 4. Please include the relative expression of the controls and indicate which of the treatments actually present statistical differences from the control treatment. Please replace the “3 days and 6 days” labels with “3 spray applications and 6 spray applications” if this is the case.

done

24- Line 126. Please remove “differential”.

done

25- Lines 139-140. Please rephrase “positive relation was observed between the molecular performance and plant behavior in controlling tomato black mold disease” with “the induction of defense-related genes could partially explain the enhanced control of tomato black mold disease” or similar.

done

26- Lines 177-178. Please replace “This resistance was induced through the expression of the seven target defense genes” by “This resistance was accompanied by the augmented expression of seven defense-associated genes (RAP, XET–2, ACS–2PINII, PAL5, LOXD, and PR1) probably showing the induction of SAR” or similar.

done

27- Lines 179-183. The last two sentences are almost the same, please rephrase. On the other hand, I think any producer will select the chemical based on gene expression profiles, so I suggest selecting the best inducer based on overall productivity and quality.

rephrased

"Authors’ response to reviewers’ comments"

Gene Expression Signals of Some Safe Inducers Used to Control Tomato Black Mold Disease, and Enhance Productivity and Quality

Hoda A. S. El-Garhy1, Fayz A. Abdel-Rahman2, Abdelhakeem S. Shams3,
Gamal Osman 4,5,6,* and Mahmoud M. A. Moustafa1

Comments and replies

We would like to thank the reviewer 2 for the valuable comments and constructive suggestions to our manuscript, which greatly helped to enhance the quality of our manuscript.

For the provided specific points:

we have mad the requested modifection and hope the revised version would meet the requirements.

 Reviewer 2 comments:

Authors replies

1- Please rephrase the title, especially don’t use the word “some” as it is too vague. I suggest:

Comparative analyses of four chemicals used to control Black Mold Disease in tomato and its effects on defense signaling pathways, productivity and quality traits

We accepted the suggested title and repaced the old one by it, thank you for it

2- Abstract

Please introduce the chemicals and their use before start summarizing the results. Please reduce the results showed in the abstract to highlight only the most relevant ones.

- done

3- Line 45: please rephrase with: “However, production is threatened by many different pathogens that affect quality and productivity.” Or similar.

- done

4- Line 60: Please remove “on the other hand”.

done

5- Line 65: Please rephrase “weight gain” with “biomass accumulation” or similar.

-       Done  

6- Line 67: Please add “Lithovit is a foliar fertilizer that improves plant growth

- done

7- Line 73: Please rephrase with “It has been proposed that

done

8- Lines 74-79. Please replace these examples by “Potassium (K+) is a macronutrient required for fundamental physiological and molecular processes in plants.” Or similar.

done

9- Lines 82-84. Please introduce SAR before talking about the genes. Something like: “Systemic acquired resistance (SAR) is a mechanism by which plant defenses are preconditioned by a prior treatment that results in enhanced resistance against subsequent pathogen infections. Although the molecular details of the signaling machinery are poorly understood, previous studies have shown…” or similar.

done

10- Lines 90-96. Please rephrase, you are studying the chemicals in many different contexts. Something like “This study aimed to analyze the effectiveness of chitosan, salicylic acid, lithovit, and potassium thiosulfate in improving tomato productivity and quality. We assessed if these chemicals affect fungal growth, plant growth parameters, cold storage period, yield components, and fruit quality, as well as modifying post-harvest resistance to Alternaria alternata infection. Finally, we addressed whether we could identify the induction of SAR-related defense genes by analyzing the expression of RAP, XET–2, ACS–2, PINII, PAL5, LOXD, and PR1” or similar.

done

11- Line 132. Please rephrase, as the chemicals are not acting as inducers in this assay. “The effect of the chemicals on the growth of Alternaria alternata…” or similar.

done

12- Line 245. Please add a suitable reference showing that Actin is a good reference gene in similar conditions as the ones assayed in this work.

Done, the requested reference (Upadhyay et al. 2014) was added to M&M section and to table 2

13- Line 288. Please remove “the”

done

14- Line 310. A dot is missing after “[48]”.

done

15- Lines 312-313. It is not clear what you want to say about clay soil, please rephrase. 

rephrased

16- Line 4. Please replace “evident from” with “observed in”.

done

17- Line 5. Please replace “outcome” with “fruit production”

done

18- Tables 4 and 5. I strongly suggest to the authors to replace these tables with graphical charts. This will facilitate the analysis and will allow incorporating the dispersion of the data (standard error/standard deviation), which is not incorporated in the tables. I also suggest using a graphical representation that includes individual data points as proposed in this work https://journals.plos.org/plosbiology/article?id=10.1371/journal.pbio.1002128.

done

19- Tables 6 and 7. Please try to join the data in one single table summarizing the results from both seasons. If possible, also use a graphical representation.

If we joined them the resulted table will apear more coplex, so, it will be better to keep both tables separated

20- Line 21. Please include a brief introduction about why are you studying chitinase and catalase activities.

done

21- Figures 1 and 2. Please replace the left panel named “Alternaria alternata” with “artificially inoculated” or something similar. Please also consider including the individual data points in the graphs as suggested before.

done

22- Line 83-84. Please replace “Gene Expression Signals of Resistance Genes in Treated Tomato Plants” with “mRNA expression of defense-related genes” or something similar.

done

23- Figures 3 and 4. Please include the relative expression of the controls and indicate which of the treatments actually present statistical differences from the control treatment. Please replace the “3 days and 6 days” labels with “3 spray applications and 6 spray applications” if this is the case.

done

24- Line 126. Please remove “differential”.

done

25- Lines 139-140. Please rephrase “positive relation was observed between the molecular performance and plant behavior in controlling tomato black mold disease” with “the induction of defense-related genes could partially explain the enhanced control of tomato black mold disease” or similar.

done

26- Lines 177-178. Please replace “This resistance was induced through the expression of the seven target defense genes” by “This resistance was accompanied by the augmented expression of seven defense-associated genes (RAP, XET–2, ACS–2PINII, PAL5, LOXD, and PR1) probably showing the induction of SAR” or similar.

done

27- Lines 179-183. The last two sentences are almost the same, please rephrase. On the other hand, I think any producer will select the chemical based on gene expression profiles, so I suggest selecting the best inducer based on overall productivity and quality.

rephrased